# The Use of Gaussian Mixture Models with Atmospheric Lagrangian Particle Dispersion Models for Density Estimation and Feature Identification

**Alice Crawford** [†] 

NOAA Air Resources Laboratory, College Park, MD 20740, USA; alice.crawford@noaa.gov
† Current address: 5830 University Research Court, College Park, MD 20740, USA.

**Abstract:** Atmospheric Lagrangian particle dispersion models, LPDM, simulate the dispersion of passive tracers in the atmosphere. At the most basic level, model output consists of the position of computational particles and the amount of mass they represent. In order to obtain concentration values, this information is then converted to a mass distribution via density estimation. To date, density estimation is performed with a nonparametric method so that output consists of gridded concentration data. Here we introduce the use of Gaussian mixture models, GMM, for density estimation. We compare to the histogram or bin counting method for a tracer experiment and simulation of a large volcanic ash cloud. We also demonstrate the use of the mixture model for automatic identification of features in a complex plume such as is produced by a large volcanic eruption. We conclude that use of a mixture model for density estimation and feature identification has potential to be very useful.

**Keywords:** HYSPLIT; LPDM; modeling; atmospheric; dispersion; volcanic; ash; tracer; Gaussian mixture model

## 1. Introduction

Atmospheric Lagrangian particle dispersion models, LPDM, model the path of computational particles which represent passive tracers in the atmosphere. At the most basic level, model output consists of the position of computational particles and the amount of mass they represent. In order to obtain concentration values, this information is then converted to a mass distribution via density estimation. This task is usually integrated into the modeling code itself, so that the main model output consists of gridded concentrations. The estimation of gridded concentrations from computational particle positions can be accomplished in several different ways.

One simple, common, and effective way is to use a bin counting or histogram method. A three dimensional grid, the concentration grid, is defined and gridded concentrations are calculated by summing the mass of the particles within each grid box and dividing by the volume. This method works quite well for regions in which the number of computational particles in the bin, N, is large but shot noise dominates at lower values of N. The number of particles simulated and the size of the bins must be chosen carefully to take into account the region of interest, the desired spatial and temporal resolution, and the lowest concentrations which must be resolved. In some cases it is desirable to define a high resolution concentration grid for close to the source and a coarser concentration grid for computing concentrations far from the source [1].

Another nonparametric method is the kernel density estimator, KDE. The position of each computational particle, a Dirac delta function in the histogram method, is replaced by another function or kernel, such as a Gaussian, so the mass represented by the particle is distributed in space. A

continuous concentration field is the result of summing over the individual mass distributions. Usually a single kernel is chosen to represent all particles, but the kernel bandwidth (for instance the variance for a Gaussian kernel) may be different for different computational particles. For example, in some formulations, the bandwidth of particles which have been through more turbulent regions of the atmosphere may be larger. Flexpart utilizes a uniform kernel after three hours of simulation and has an option to employ a more computationally expensive parabolic kernel the width of which depends on particle age [2]. Choosing the kernel and estimating the bandwidth can be fraught and ad hoc. Refs. [3,4] provide a good discussion. Gridded output is obtained by summing the mass within each grid. Mass from a single particle may be spread over multiple grid boxes thus when gridded to the same resolution, the use of a KDE generally produces a smoother concentration field than the histogram method. It is used in some applications as a way to reduce the number of computational particles needed for a simulation [5].

Here we introduce the use of Gaussian Mixture Models, GMM, to produce the density estimate. Mixture models are so named as they are a mix between a parametric method in which the underlying distribution assumes is assumed to assume a certain form and nonparametric methods such as the histogram and KDE methods discussed above. A GMM produces an estimate of the mass distribution which is the weighted sum of a number of Gaussians. Other mixture models may use distributions besides Gaussians. We focus on GMM's here because the computational tools to use them are easily accessible and there is ample reason to believe that a mix of Gaussians can represent the distribution adequately. For instance, Gaussians are often used as the kernel shape in the KDE and Lagrangian puff models use Gaussian "puffs" which travel with the mean wind and simulate turbulent dispersion by expanding [6,7].

Using a GMM has some striking advantages for certain situations. Use of the GMM can significantly reduce the number of computational particles needed for a simulation which results in significantly lower computational costs. These models are often used in emergency response applications in which require fast response times. The resulting mass distribution can be prescribed concisely by specifying the parameters of a limited number of Gaussians. This could lead to more compact ways of storing and transmitting data as well as combining and comparing results from dispersion ensembles.

GMM's are also used as clustering algorithms and the output can be used to automate identification of features in the concentration field. For instance, volcanic clouds often develop into complex three dimensional structures and feature identification can aid in data assimilation procedures, identifying areas of uncertainty in the forecast cloud and computing object based verification statistics.

In the following sections we first use a tracer experiment to validate our GMM approach to density reconstruction and compare it to the histogram method. We then provide examples of its use for modeling distal volcanic ash clouds.

## 2. Method

### 2.1. Tracer Data

The CAPTEX data consists of six 3 h duration releases of perfluoromonomethylcyclohexane (PMCH) near the surface [8,9]. The releases took place between 18 September and 29 October 1983. Releases 1–4 were from Dayton, Ohio. Releases 5 and 7 were from Sudbary, Ontario, Canada. 3 h and 6 h averaged air samples were collected at 84 measurement sites distributed from 300 to 800 km downwind of the release site for up to 60 h after the release. Captex 1–4 releases occurred in the afternoon during well mixed conditions, while Captex 5 and 7 occurred at night after the passage of a front associated with strong wind conditions. Further discussion of meteorological conditions during the release can be found in [10].

The ranking method of [11] is used to evaluate performance of the simulations. The rank combines the normalized correlation coefficient *R*, fractional bias FB, figure of merit in space FMS, and

Kolmogorov–Smirnov parameter KS, into a single parameter which ranges from 0 to 4 with 4 being the best score.

$$\text{Rank} = R^2 + (1 - |\text{FB}/2|) + \text{FMS}/100 + (1 - \text{KS}/100) \tag{1}$$

Fractional bias is a normalized measure of bias which ranges from $-2$ to 2 [12]. The FMS is sometimes called the critical success index, CSI or threat score [12,13] and it is the ratio of the area of overlap between the measurements and predictions to the area of nonoverlap multiplied by 100. KS is defined as the maximum difference between the measured and predicted cumulative distribution functions (CDF) [13].

### 2.2. Lagrangian Atmospheric Transport and Dispersion Model

The Hybrid Single Particle Lagrangian Integrated Trajectory model, HYSPLIT, v.5.0.0 was used [7] for all simulations, although the technique could be used with any LPDM. Defaults for HYSPLIT were used except where noted.

### 2.3. Simulations of CAPTEX

For the CAPTEX tracer experiment simulations, defaults for HYSPLIT v5.0.0 were utilized except the model time step was set at 5 min. HYSPLIT was driven by the WRF dataset described in [14] which has 27 km horizontal resolution and 1 h temporal resolution. Table 1 gives information about the five different simulation setups that were used to simulate the 6 CAPTEX releases. The number of particles released per hour was varied, as was the concentration grid which defines the three dimensional grid for the histogram method of density reconstruction. The standard run grid resolution and number of particles has been used in past studies [7,14]. Further information on the simulations can be found in the Supplementary Materials.

**Table 1.** Summary of HYSPLIT simulations. RunC and RunD are identical except for the random seed which was used in the simulation. $C_l$ is given in Equation (2). For the CAPTEX simulations, the number of particles is the number of particles released over the 3 h emission period. For the Kasatochi simulations the number of particles is approximately the total number of particles released over 8 h. The number of particles changed due to deposition.

| CAPTEX Simulations | | | | | |
|---|---|---|---|---|---|
| **RunID** | **Number of Particles** | **Horizontal Resolution** | **Vertical Resolution** | **Note** | $C_\ell$ **pg m**$^{-3}$ |
| A | 250,000 | $0.05° \times 0.05°$ | 25 m | | 39 |
| B | 50,000 | $0.05° \times 0.05°$ | 25 m | | 196 |
| C | 5000 | $0.05° \times 0.05°$ | 25 m | | 1958 |
| D | 5000 | $0.05° \times 0.05°$ | 25 m | SEED = $-4$ | 1958 |
| E | 50,000 | $0.25° \times 0.25°$ | 100 m | Standard run | 2 |
| Kasatochi Simulations | | | | | **mg m**$^{-3}$ |
| KA | 500,000 | $0.25° \times 0.25°$ | 1 km | Reference run | 0.08 |
| KB | 53,000 | $0.25° \times 0.25°$ | 1 km | | 0.75 |
| KC | 26,000 | $0.25° \times 0.25°$ | 1 km | SEED = $-4$ | 1.5 |
| KD | 26,000 | $0.25° \times 0.25°$ | 1 km | SEED = $-6$ | 1.5 |

### 2.4. Histogram Method

It is instructive to calculate the lowest concentration that can be simulated using the histogram method. This is the concentration that would be calculated from one computational particle spending one time step in a bin.

$$C_\ell = \frac{\dot{m}}{\dot{p}V} \frac{\Delta t}{T_{\text{ave}}} \tag{2}$$

where $\dot{m}$ is the mass released per unit time, $\dot{p}$ is number of computational particles released per unit time, $V$ is the volume of the bin, $\Delta t$ is the model time step, and $T_{\text{ave}}$ is the averaging time. The model time step defines the integration time step and determines how often other simulation tasks, such as summing the mass in the concentration grid, or emitting new computational particles are performed. $C_{\ell}$ for the CAPTEX 1 simulation for a 3 h averaging time and model time step of 5 min is given in Table 1. Release rates for the six captex experiments were $6.9333 \times 10^4$, $6.7 \times 10^4$, $6.7 \times 10^4$, $6.6333 \times 10^4$, $6.0 \times 10^4$, and $6.1 \times 10^4$ g h$^{-1}$ for experiments 1,2,3,4,5,7 respectively.

There is some uncertainty in concentrations determined by the histogram due to shot noise. The probability distribution function, PDF, of the number of computational particles found in a certain volume is well represented by the Poisson distribution (Appendix A). If we take the standard deviation to be a good measure of the uncertainty in the number of particles in the volume, then a well known results is that the relative or normalized uncertainty in the concentration is given by

$$\frac{u_c}{C_M} = \frac{1}{\sqrt{N_M}}$$

Then the absolute uncertainty, $u_c$, in the concentration due to shot noise is a function of the concentration and can be related to $C_{\ell}$ in the following way.

$$N_M = \frac{C_M}{C_{\ell}} \tag{3}$$

$$u_c(C_M) = C_{\ell}\sqrt{N_M} = \sqrt{C_M C_{\ell}} \tag{4}$$

where $N_M$ is the mean (and variance) of the Poisson distribution and $C_M$ is the concentration that would result from having $N_M$ particles in the volume. This assumes all particles have the same amount of mass and their mass does not change over time.

In the CAPTEX 1 experiment, the lowest measured concentrations were on order of 10 pg m$^{-3}$ and the standard run, runE, is configured so that $C_{\ell} = 2$ pg m$^{-3}$ and $u_c(10) = 4.5$ pg m$^{-3}$ (given $T_{\text{ave}} = 3$ h and $\Delta t = 5$ m). The combinations of particle number and concentration grid resolution in the other runs are designed to be suboptimal.

*2.5. Simulations of Volcanic Eruption*

Here we follow [15] in simulating the eruption of Kasatochi in the Aleutian Islands in 2008. The volcanic ash simulations use the same cylindrical source term as in [15]. Mass is distributed uniformly in a cylindrical shape 100 m in diameter from the volcano vent at 0.3 km to 18 km. A unit mass is released per hour starting from 8 August 2008 04 UTC and continuing for 8 h. The mass eruption rate, $\dot{m}$, of a volcanic eruption is often estimated using an empirical relationship which is a function of plume height [16]. For aviation forecasts, this number is further reduced by up to 10% to provide a release rate of fine ash. Because of the uncertainty in the estimation, volcanic ash simulations are often run releasing an arbitrary unit of mass every hour. Concentrations in the arbitrary unit of mass per m$^{-3}$ are then converted to actual concentrations in postprocessing. Here we run emitting a unit mass per hour and then apply a mass eruption rate of fine ash of $\dot{m} = 2.8 \times 10^5$ kg s$^{-1}$. This is larger than [15,17] suggest was the actual eruption rate but smaller than the rate calculated from the relationship in [16]. To see the effect of reducing $\dot{m}$ by a factor of ten, simply divide all mass loadings and concentrations in the figures by 10.

Table 1 provides a summary of the runs used. They differ only in the number of particles and random seed. The reference run, KA, is so-called simply because it utilizes the most particles and so should provide low noise concentration estimates in the main part of the plume.

Gravitational settling and dry deposition was included in the volcanic ash simulations (wet deposition was not). The default in HYSPLIT is for particles within the deposition layer to lose a percentage of their mass to deposition. However, as will be discussed in Section 2.6, the current

implementation of the GMM cannot handle computational particles with changing mass. In order to keep the mass on the particles the same, we used the option to instead remove a percentage of computational particles within the deposition layer. Thus the number of particles changed over the simulation period, but the mass on each computational particle remained the same throughout. A time step of 10 min was used for these simulations. For this study we ignored particles in the first level (0–1 km) in order to focus on the higher level ash.

One ensemble member of the ERA5 ensemble [18] was used to drive HYSPLIT. The data has 3 h temporal resolution and approximately 70 km spatial resolution.

For simplicity we look at output from one time step, rather than an average as we did for the CAPTEX tracer experiment. Typically volcanic ash simulations utilize a 1 h averaging time. The use of the time average decreases $C_\ell$ and, more importantly, may help account for uncertainty in timing. However, model output consisting of snapshots such as shown here may be desirable in some cases such as when assimilating satellite data which may be available every 10 min into an inversion algorithm.

Four particle sizes are specified for the simulation. Particles with radius of 10, 3, 1, and 0.3 µm make up 67%, 25.4%, 6.8%, 0.8% of the mass respectively. This is the current default particle size distribution used when HYSPLIT is run operationally at the volcanic ash advisory centers, VAACs. HYSPLIT splits the number of computational particles released evenly among the number of species defined so computational particles of the largest size, 10 µm carry approximately 2.6, 9 and 84 times as much mass as the particles of size 3.1 and 0.3 µm respectively. A $C_\ell$ in this case can be computed separately for each particle size. Care should be taken because the mixture of computational particles with vastly different masses may make it appear that the particle number and grid resolution are adequately matched when they are not. The largest value should generally be used when determining an appropriate concentration grid size. The values in Table 1 are for the 10 µm size particles at a latitude around 45°, assuming $\dot{m} = 2.8 \times 10^5$ kg s$^{-1}$ and no time averaging, that is $\Delta t = T_{\text{ave}}$.

For the density reconstruction with the GMM, each particle size is fit independently. Total concentrations can be found by summing the concentrations from each particle size.

*2.6. Gaussian Mixture Model*

The scikit-learn [19] Python package was used to fit the mixture models to the HYSPLIT output. Here we employ the Gaussian mixture model (GMM) and Bayesian Gaussian Mixture model (BGMM) classes provided in the scikit-learn module.

The PDF for a GMM is

$$f(x) = \sum_{m=1}^{n} \sigma_m \phi(x; \mu_m, E_m) \tag{5}$$

where $n$ is the number of Gaussians in the model, $\sigma_m$, $mu_m$ and $E_m$ are the weight, mean and covariance matrix of the mth Gaussian and $\phi(x)$ is the form of the Gaussian distribution [20]. The parameters for the GMM will be referred to as $\theta$ and consist of the set of $\sigma$, $\mu$, and $E$.

The task of the Gaussian mixture model class within the scikit-learn Python package is to find the best $\theta$ for a given $n$ and set of points. Here, the points are the locations of the computational particles. The resulting PDF represents the probability of finding a computational particle at a particular location. The task is accomplished by using an expectation maximization, EM, algorithm to find a maximum in the log-likelihood function, $L(\theta)$ which is simply the sum of the log of the PDF evaluated at each point [19,20].

$$L(\theta) = \sum_{i=1}^{P} \log f_\theta(x_i) \tag{6}$$

One criticism of the EM method is that it can be sensitive to the initialization as it may return a local maximum, of which $L(\theta)$ may have many. Still, the EM method remains quite popular and effective possibly because often the local maximum is good enough. The scikit-learn package does

allow the user to specify the number of initializations to be performed and will return the result with the highest $L(\theta)$. Here we used the default which is one initialization which is obtained from a kmeans clustering of the data [19]. Note that there are other methods for fitting a GMM [21–23] some of which may be more efficient, less sensitive to initialization, or have additional benefits such as estimating n. The fitting time generally increases as P and n increase. Ref. [21] states that the time for their algorithm to complete increases linearly with P and quadratically with n. Note that the computational time for a LPDM simulation to complete also increases linearly with P. A rigorous evaluation of the computational time is outside the scope of this paper; however, subjectively we found the fitting time for the simulations with low and medium numbers of particles to be reasonably fast.

For the examples discussed here the position of the computational particles was specified as degree longitude and latitude for the x and y coordinate and kilometers above ground level for the z coordinate. This choice does affect the fits, as it causes a vertical spacing between points of 1 km to appear approximately the same as a horizontal spacing of 100 km to the algorithm. This ostensible compression of the horizontal scale places more emphasis on horizontal groupings of particles which we subjectively felt was desirable for feature identification. The density estimation is not sensitive to this choice.

This package does not allow points to be weighted, so all particles being fit have the same mass. When particles with different masses are present, each set of particles may be fit separately and a final concentration determined by summing the contributions from each. In the future, an algorithm which allows the use of weighted points may be used [24]. For example, this would be important for applications in which radioactive decay or some chemical transformation is significant.

When fitting distributions which should be similar (e.g., different sized particles at the same time, particles at fairly closely spaced times, output from different dispersion ensemble member runs), the fit from one distribution may be used to initialize the next fit. This can decrease computational time. The probability density is transformed into a mass density simply by multiplying by the total mass of all the particles. Concentration in a volume is obtained by integrating the PDF over the volume in question, multiplying by the total mass, and dividing by the volume. Integrating over the fits to obtain the gridded data also incurs a computational cost. In the current implementation this step, which was not optimized, tended to take longer than the fitting. It is expected that the speed can be improved. In addition, there may be cases in which it is not desirable to obtain the full concentration field but only retrieve concentrations at certain locations such as those corresponding to measurement points or locate regions of highest concentration.

Time averaging may be accomplished in two ways. A fit may be applied to all positions which occur within the averaging time. In the current implementation this only works if particle positions are output at regular intervals (e.g., the time step in HYSPLIT is constant). If an algorithm that accepts weighted points is implemented, then variable time steps could be used. Another method is to create a fit for particle positions at each output time within the averaging time and then average the results of all the fits. These two methods are illustrated in Figure 1. The first method is more computationally efficient as less fits need to be performed and then integrated over. The fitting algorithm does not recognize the existence of hard boundaries such as the ground and thus some mass may be spread underground. Here we employ the simple method of redistributing mass below ground level into the first level. Reflecting the mass below ground as [3] does for KDE would produce similar results. Other methods which may be investigated in the future include employing the GMM only in two dimensions near the ground and assuming a uniform vertical distribution of the computational particles and employing a mixture model that uses different distributions.

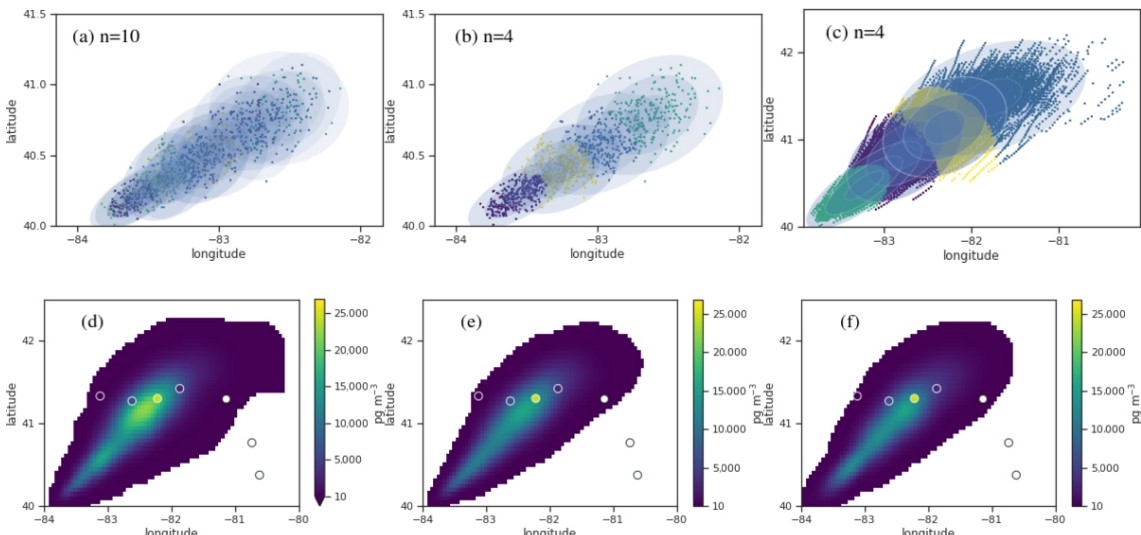

**Figure 1.** The top panels, (**a**–**c**) show particle positions at (**a**,**b**) 09/18/1983 21:00 UTC and (**c**) 09/18/1983 from 21 to 24 UTC with particle positions output every 5 min. The Gaussian fits are shown in light blue and the colors of the particles indicate which cluster the particles were assigned to by the algorithm. The bottom panels, (**d**–**f**) show 3 h averaged concentrations calculated from the fits on a $0.05° \times 0.05°$ 25 m grid from the ground to 25 m. A threshold of 1 pg m$^3$ has been applied and station measurements for the CAPTEX1 experiment are shown by the circles. (**d**,**e**) were calculated from separate fits to each time period and the panels above show the first such fit. (**f**) was calculated from one fit to all time periods which is shown in (**c**).

The scipy module defines a score, $S_{ij}$ which is the per-sample average log-likelihood.

$$S_{ij} = \frac{1}{P}L(\theta_i, p_j) \tag{7}$$

where $\theta_i$ are parameters resulting from fitting to a set of points, $p_i$, and $L(\theta_i)$ is evaluated over a set of points, $p_j$, which may or may not be the same as $p_i$. $P$ is the number of points in $p_j$. The score indicates how well the PDF describes the set of points and is best used in a relative sense, to compare different fits to the same set of points, or different sets of points to the same fit.

Generally, increasing n, will increase a score when $i = j$. Trying to determine n by maximizing $S_{ii}$ will likely result in overfitting. However, a possible method for helping determine n is to examine $S_{12}$ and $S_{21}$ from two simulations which are expected to produce the same distribution (e.g., two identical runs with different random seeds). This is discussed more in Appendix B.

The scipy module contains functions for calculating the Akaike information criterion (AIC) and Bayesian information criterion (BIC) which can be used to help determine the optimal number of Gaussians to use [20]. Presumably the optimal n produces the lowest AIC or BIC which is a balance between finding a low $S_{ii}$ and keeping n small. We did not rely on these criteria for the results presented here as we found some overfitting was desirable for the density reconstruction.

## 3. Results

### 3.1. Density Reconstruction for a Tracer Experiment

Figure 2 compares a cross section of the modeled plume from runs A,B,C,D,E using the histogram method and run A,B,D using the GMM approach. The GMM is able to construct a concentration field at high spatial resolution even with very low number of particles. Comparing Figure 2f with (i) and (g) with (j) shows that there is not much difference between using the two methods of time averaging. Figure 2h suffers slightly from overfitting. Two maxima are observed where presumably there should

only be one and the edges defined by the 10 pg m$^{-3}$ threshold are oddly lumpy. Still, the results are preferable to the histogram counterpart shown in Figure 2c and reducing, n, as shown in (k) remedies the overfitting.

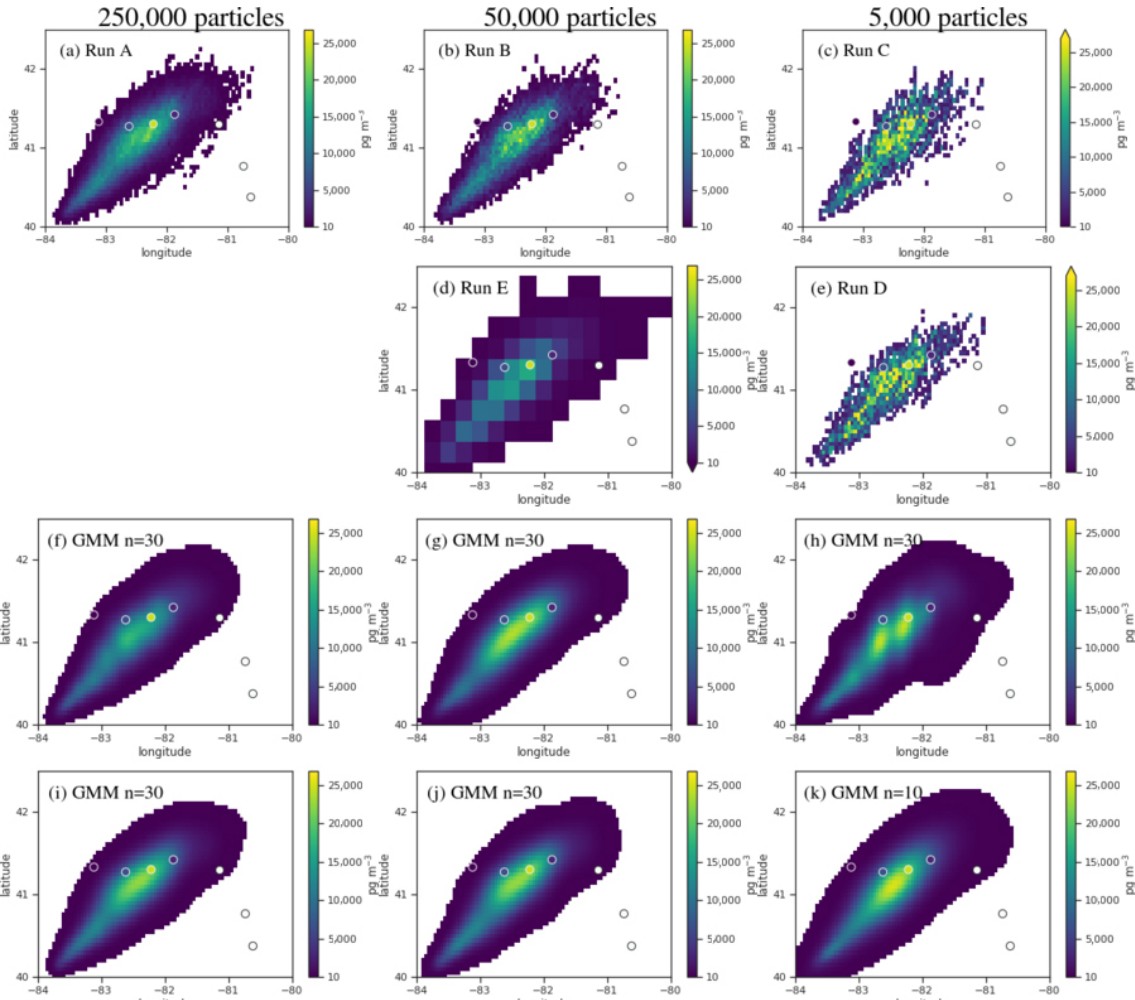

**Figure 2.** Comparison of density estimations. Color shows concentrations in pg m$^{-3}$. Shown is a cross section of the simulated plume from the CAPTEX 1 experiment with station measurements shown by the circles. Simulated concentrations are 3 h averages on 09/18/1983 from 21 to 24 UTC, 3 h after the release. all figures have horizontal resolution of 0.05$^{\circ}$ × 0.05$^{\circ}$ × 25 m except for the control runE, (**d**) has resolution of 0.25$^{\circ}$ × 0.25$^{\circ}$ ×100 m. Number of particles released per hour is shown at the top of each column. (**a**–**e**) are calculated using histogram method. (**f**–**k**) are calculated using a GMM with number of components annotated on the graph. Particles from the ground to 500 m were used in the fit. A threshold of 10 pg m$^{-3}$ was applied to (**f**–**k**). (**i**,**j**) were calculated from fits to each time step in the averaging period, while (**f**,**g**,**h**,**k**) were calculated from a fit to all particles in the averaging period. (**h**) and (**k**) were calculated from Run C.

Figure 3 shows statistical rank for the six CAPTEX releases, the statistics which make up rank, and the root mean square error, RMSE. The plots are configured so that performance of the histogram method with a high resolution concentration grid can be more easily compared to the standard run and the GMM method.

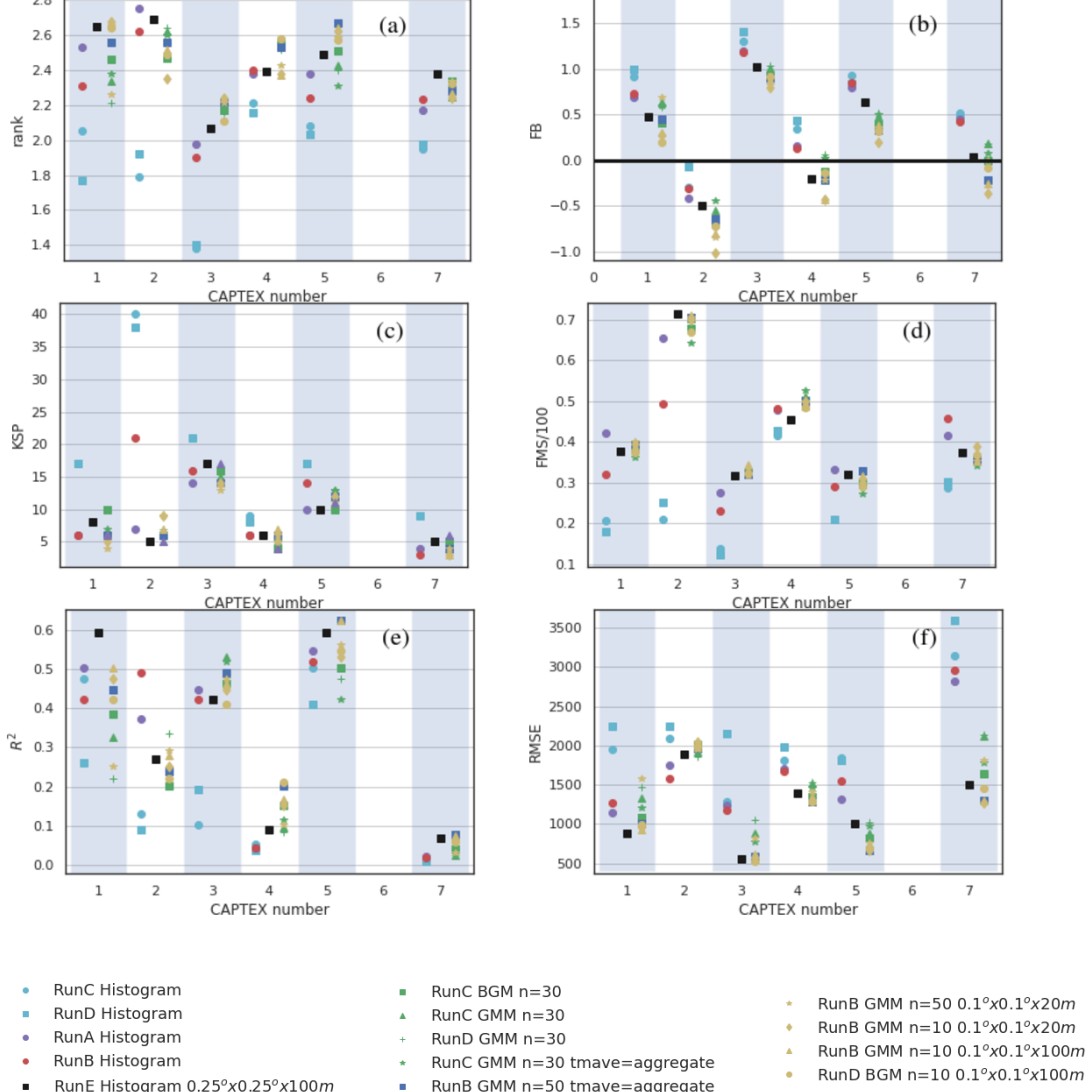

**Figure 3.** Statitics for CAPTEX experiments using a variety of density reconstructions. The rank, described by Equation (1) is shown in panel (**a**). The four statistics which go into calculating rank are shown in panels (**b**–**e**) and the root mean square error, RMSE, is shown in (**f**). Values for the standard simulation, runE, with standard concentration grid are shown by the black square. Values for concentrations calculated with the histogram and a high resolution grid are plotted slightly to the left of the black square. Values for concentrations calculated with the mixture model are plotted to the right of the black square. The high resolution grid of $0.05° \times 0.05° \times 25$ m was used unless otherwise noted in the legend. For the GMM, the time averaging was performed with a separate fit to each output time unless tmave = aggregate is noted in the legend.

The main result is that simulations using only 5000 particles and the GMM performed as well, as measured by rank, as simulations using 50,000 and 250,000 particles and the histogram method to produce concentrations on a high resolution grid. In the histogram method, there was a significant improvement when using 50,000 particles over 5000 particles. The performance of the 250,000 particles and 50,000 particles was similar, although the 250,000 particle simulation generally performed slightly better.

One apparent trend is that the FB for concentrations determined from the GMM density estimation are lower than all of the high spatial resolution histogram runs (Runs A,B,C,D). They tended to be

slightly lower than the standard Run E as well. This is advantageous for CAPTEX 1,3,5, and 7 and disadvantageous for CAPTEX 2. The higher FB for Runs A,B,C,D histogram method is a result of the relatively high value of $C_\ell$ and the fact that the Poisson distribution is positively skewed.

The performance of the GMM density reconstruction did not exhibit much dependence on choice of n (number of Gaussians to be fit), particle number, output grid resolution, or time averaging method. The FMS and KSP for all GMM output are tightly clustered and close to the value for the standard run. The most variation is seen in $R^2$ but it is difficult to detect a trend.

### 3.2. Density Reconstruction for a Volcanic Eruption

This section mainly comprises discussion of Figures 4–6. In each of these, concentrations or column mass loadings derived using the histogram method are shown in the top row and the reference run, KA, is always shown in the uppermost left corner.

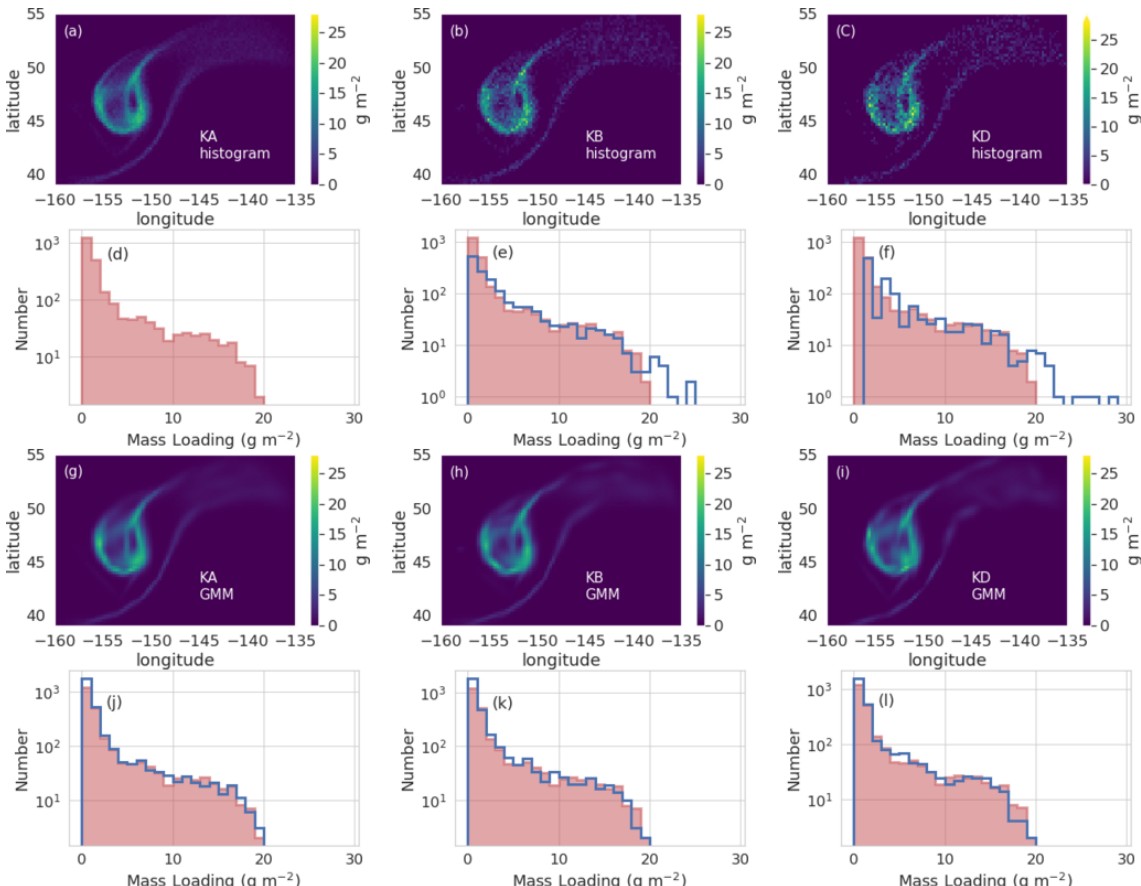

**Figure 4.** Mass loading from simulations of the 2008 eruption of Kasatochi. The first and third rows (**a–c,f,h,i**) display plots of mass loading with the simulation used and method of density reconstruction noted in the bottom right corner. The GMM used 50 Gaussians and output displayed at the same resolution as the histogram plots ($0.25° \times 0.25°$ grid). Directly below each mass loading plot is a histogram of the mass loading values with a threshold of $0.01$ g m$^{-2}$ applied. The histogram for (**a**) is displayed as the red shaded region in each histogram plot for easy comparison.

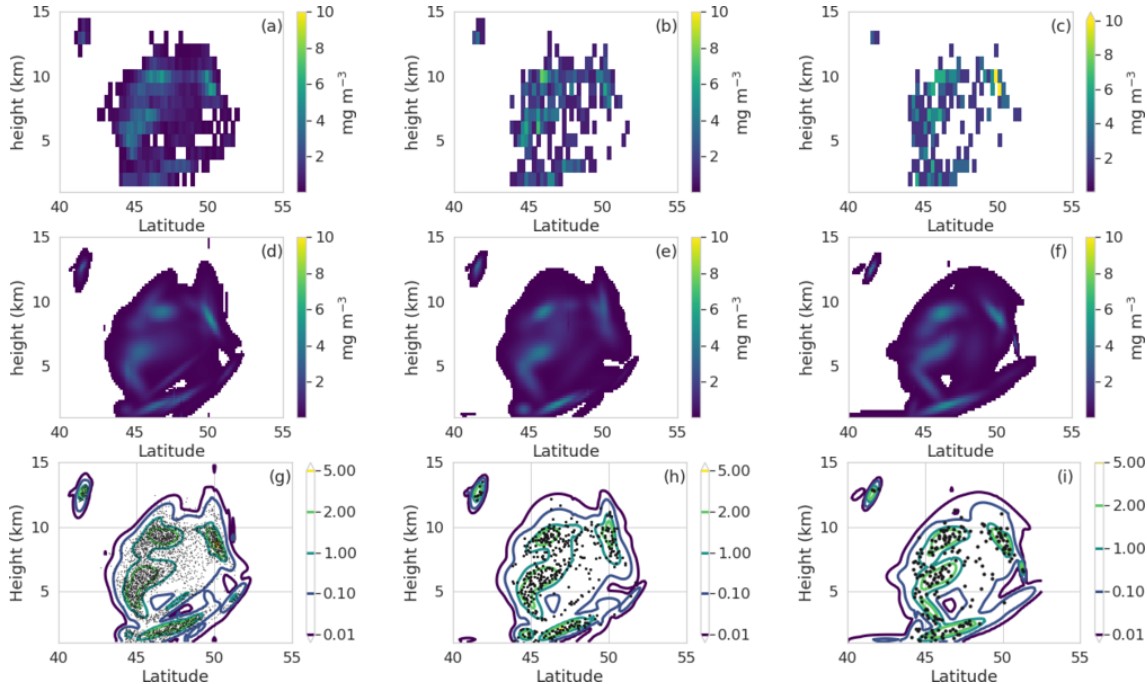

**Figure 5.** Concentrations at longitude-151. Top row (**a**–**c**) concentrations calculated using histogram method resolution $0.25° \times 0.25° \times 1000$ m. Second row (**d**–**f**) and third (**g**–**i**) row concentrations calculated using GMM with 50 Gaussians and plotted at resolution of $0.1° \times 0.1° \times 200$ m. For the second row concentrations below below $0.01$ g m$^{-3}$ are not shown. The third row shows contour levels of output shown in second row and the black dots indicate position of computational particles within $0.125°$ of the $-151$ line of longitude. The dots are shown smaller in (**g**) for clarity. First column (**a**,**d**,**g**) Run KA, second column (**b**,**e**,**h**) Run KB and third column (**c**,**f**,**i**) run KD.

Figure 4 depicts the column mass loading and histograms of column mass loading values for three different simulations, KA, KB, KD, using two different density reconstructions. Shot noise is readily apparent in the column mass loading plots in (b) and (c). Using a smaller number of particles results in spurious higher values in the mass loading due to the shot noise. The run with the smallest number of particles is unable even to predict concentration values lower than about $1$ g m$^{-3}$ and this results in the histogram of the mass loading values being shifted to the right from the reference run (Figure 4f).

In contrast, the density reconstruction with the GMM (Figure 4g–i) reproduces the values of the reference run extremely well. The values of mass loading produced are almost identical as evidenced by the matching histograms and the placements are quite similar as well.

The utility of the GMM density reconstruction is even more apparent when examining the three dimensional concentration field. Figure 5 shows a cross section through the thickest part of the ash cloud which contains some of the highest mass loadings while Figure 6 shows a cross section through a thin but extensive area of ash with low mass loading. Shot noise is again readily apparent in (b) and (c) of both figures. In fact shot noise is also apparent in the reference run (a) and is an inevitable part of the histogram method of density reconstruction. For simulations of sufficient length, there will inevitably be areas in which the number of particles is very small. This is not of great consequence as long as those areas are below the concentration threshold of interest. The GMM reconstructions in Figure 5 were plotted at a high resolution to emphasize that the method is well suited for applications desiring high spatial resolution output.

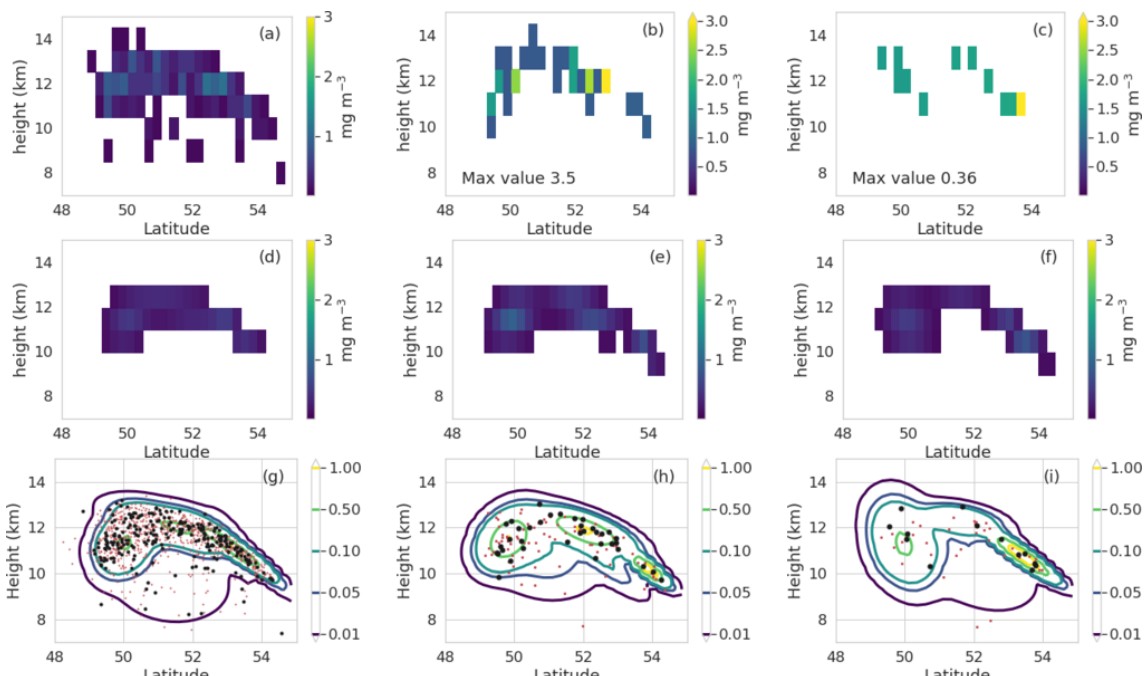

**Figure 6.** Concentrations at longitude −145. Top row (**a**–**c**) concentrations calculated using histogram method. Second row (**d**–**f**) and third (**g**–**i**) row concentrations calculated using GMM with 50 Gaussians and plotted at the same resolution as the histogram method in the second row and the contours in the third row are for with a higher vertical resolution (0.25° × 0.25° × 100 m). First column (**a**,**d**,**g**) Run KA, second column (**b**,**e**,**h**) Run KB and third column (**c**,**f**,**i**) run KD. The black dots in the third row indicate position of particles within 0.125° of the −145 line of longitude. The red dots indicate position of particles within 0.5° of the −145 line of longitude.

The similarities and differences between the three GMM reconstructions (d–i) are also more apparent on these plots than the mass loading plots. The contour plots in (g–i) in particular show that the method effectively identifies the correct general area and magnitude of the highest and lowest concentrations even with very few particles. However, there is some difference in the details.

Here we show the GMM fit using 50 Gaussians for run KA, KB, and KD. However, using 30 or 70 Gaussians produced similar results with differences akin to what you see between the three fits. Appendix B provides some further discussion and investigation of n using $S_{ij}$. Cluster assignment is more sensitive to the number of Gaussians used in the fit than the density reconstruction. This will be discussed more in the next section but overfitting, that is using more Guassians than one would use for feature identification, seemed appropriate for density reconstruction. Future work will include identifying the optimal number of Gaussians to be used in the fit. However, here we simply conclude that method is not particularly sensitive to this choice.

### 3.3. Feature Identification

In this section a few additional tasks which may be accomplished with the GMM and LPDM are introduced. The descriptions are meant to be cursory and further development of these concepts will be left as future work.

#### 3.3.1. Object Based Statistics

Object based statistics are often an attractive alternative to point based statistics. One way to define an object is with a contour line [25]. As can be seen in Figures 5 and 6, the contour lines from the GMM density reconstruction are well defined and smooth at a large range of levels and capture the overall shape and density of the particle distribution even with very few particles. The corresponding contour lines deduced directly from the histogram method (not shown) are, in contrast,

largely unintelligible for runKB and KD and even for runKA only capture the outline of the highest density areas well.

### 3.3.2. Identify Where and When Simulations Diverge

There are many cases in which simulations start with computational particles in the same initial positions, and then over time, the positions diverge. For instance, two simulations may be part of an ensemble and each member is driven with different NWP input or use different model parameterizations. Here we show how the use of the GMM provides a simple way to identify the time of divergence and the locations where the simulations differ.

Four different particle sizes were simulated for run KD with mass distributed as described in Section 2.5. The problem of specifying a particle grain size distribution, GSD, for simulations of volcanic ash is well known [26,27]. Use of an inversion algorithm for determining a GSD is challenging because separation of the particle sizes usually occurs slowly [28] and for some types of inversion algorithms adding in a GSD component would significantly increase computational costs. Considering that specification of the GSD is likely to remain uncertain, techniques to illuminate how the uncertainty in the GSD affects the forecast are needed.

Here we call $p_1$ the locations of computational particles with radius 0.3 μm, and $p_4$ the location of computational particles with radius 10 μm. The differences between the smallest three particle sizes is small and so we limit the discussion here to differences between the smallest and largest particle size.

Figure 7 shows the score as a function of time for the very smallest, $p_1$, and very largest, $p_4$, particle sizes. When $S_{14} \approx S_{41} \approx S_{44} \approx S_{11}$, the spatial distribution of the different sized particles are very similar. At around 9 August, 00:00 UTC, $S_{14}$ and $S_{41}$ start to diverge and become larger than $S_{11}$ and $S_{44}$. As time progresses, $S_{14}$ which describes how well the fit created with the small particle positions fits the large particle positions, remains greater than $S_{41}$. This indicates that regions with only small particles are generally larger and/or more distant from the overlap region than regions with only large particles. This can be easily visualized by plotting the log probability of each point in $p_i$ given the fit $f_j$. These plots are shown at two different time periods in Figure 8 for $S_{41}$, the position of the small particle size evaluated with the fit to the large particle size and $S_{14}$, the position of the large particle size evaluated with the fit to the small particle size. The first time period on 9 August at 4 UTC occurs just after the scores shown in Figure 7 start diverging. The probability for each particle can be used as a measure of how far it is from area defined by the given fit. At this time period, the separation of the largest and smallest particle sizes due to gravitational settling is evident and manifests as the smaller particles remaining at higher altitudes. In the top right panel, lower scores are evident for particles at the lower edge of the cloud. While in the panel below, the opposite is seen; lower scores are evident for particles at the top edge of the cloud.

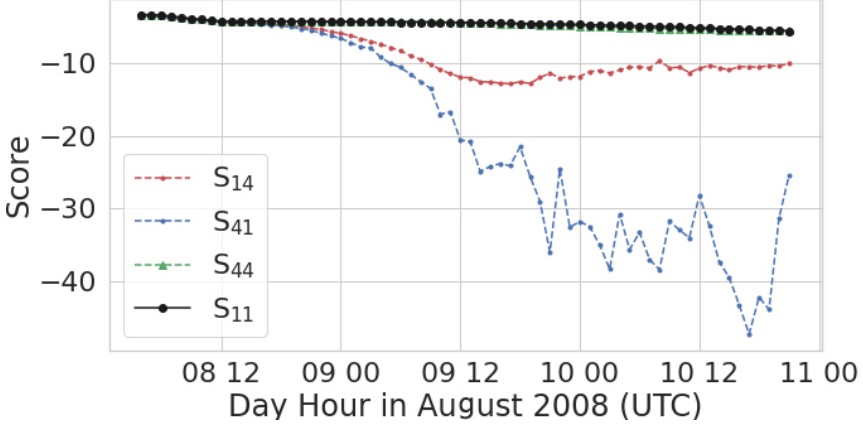

**Figure 7.** The score, $S_{ij}$, as a function of time. The score is described in the text.

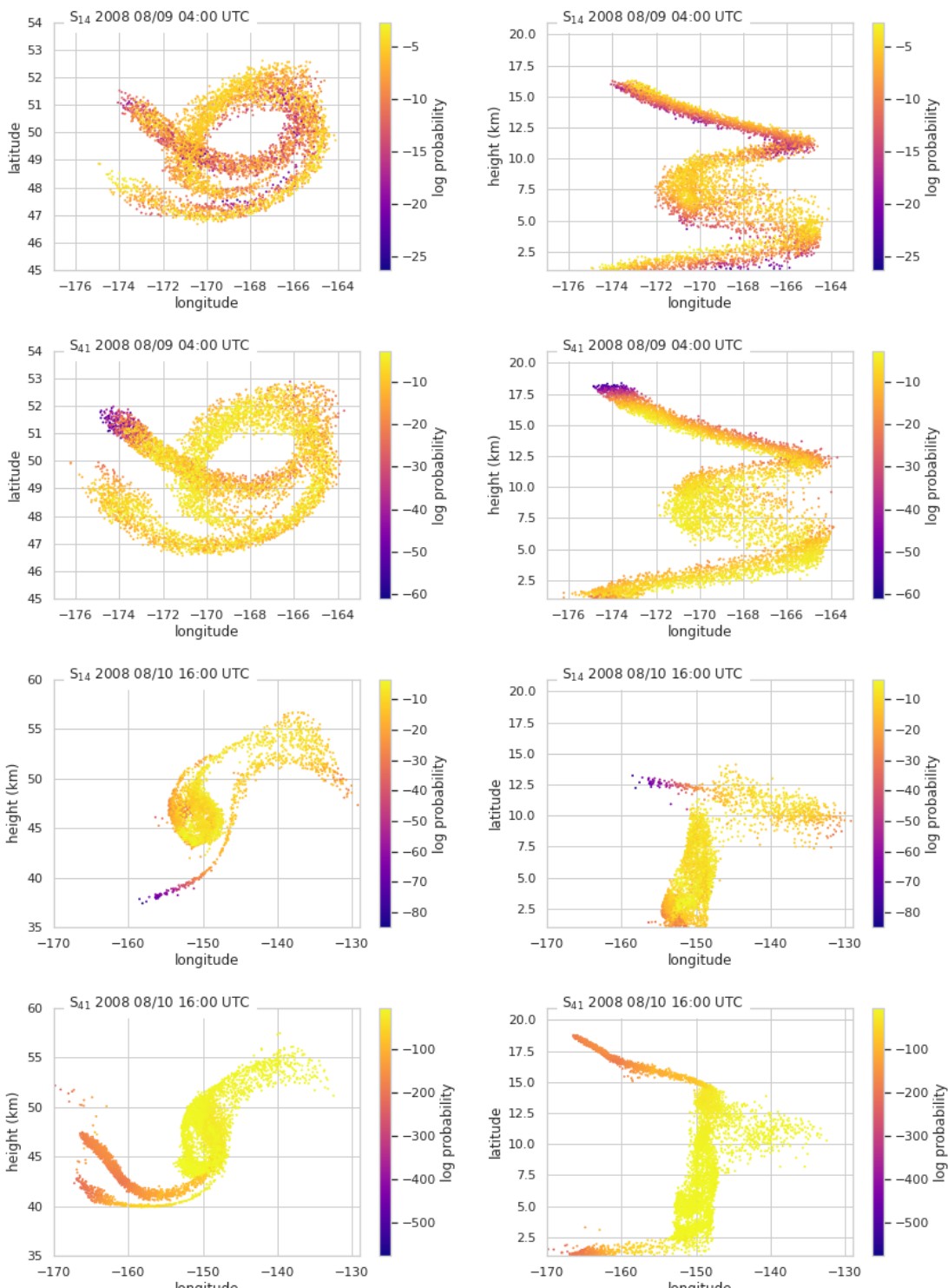

**Figure 8.** log probability of each particle position. The left column shows a projection onto the latitude, longitude plane and the right column shows a projection onto the height, longitude plane. The date is noted at the top of each figure. $S_{ij}$ indicates that the probabilities were calculated using the particle positions $j$ and the fit to particle positions $i$. For example, $S_{41}$ shows positions of the smallest particle size with how probable it is that they could belong to a fit to the largest particle size. Note the differing color scales. Low values indicate poor fit and regions where the overlap between the location of the two particle sizes is low.

Over time, this gradual vertical separation due to gravitational settling can turn into large differences in the location of different particle sizes mainly because the movement to different heights causes particles to encounter different regions of the flow [29–31]. The amount of time for significant separation to occur is expected to vary significantly for different eruptions because it depends on the details of the atmospheric flow. For the Kasatochi eruption, most of the ash was swept into an area of low pressure and we see a thick column consisting of all particle sizes traveling east along with the low.

The bottom four panels of Figure 8 show particle positions of the large particle size, $S_{14}$, and small particle size, $S_{14}$ on 10 August at 16 UTC. A significant number of small particles remained high enough that they were not swept into the center of low pressure, but instead formed a high altitude, "tail" which trails behind the main body of ash and contains none of the largest particles. A subset of the large particles encountered an attracting structure [29] which manifests as a thin line of ash around 12.5 km in height that begins south of the main column of ash. The vertically thin extrusion of ash to the north and east contains both particle sizes, but the large particles extend much further east. The small particles also form a trailing tail of ash below 2.5 km. The large particles in that area have already dropped below 1 km.

Concentrations will have less uncertainty at times for which the particle size separation is small. As more particle size separation occurs, uncertainties in modeled concentrations will also increase, particularly in areas dominated by a certain range of grain sizes. For this simulation 67% of the mass was relegated to the largest particle size, $p_4$. If more of that mass was placed in the smaller particle sizes, then concentrations in the darker colored (low probability) areas in Figure 8 $S_{41}$ could be significantly higher. If the eruption was even more coarse grained than the GSD suggests, those areas of ash could be essentially nonexistent. On the other hand, more confidence can be assigned to concentrations in areas in which all particle sizes are found, the light colored regions in all panels of Figure 8.

Identifying when, where, and how much simulations diverge can be accomplished in many ways and is related to model forecast verification. There are a number of spatial forecast verification methods in use for dispersion model forecasts [32–35] and any of the measures could be tracked over time to produce similar information to that found in Figure 7. For example, ref. [30] tracks the temporal evolution of the fractional skill score, FSS, for a volcanic ash simulation to identify when the model forecast diverges significantly from observations. Measures of separation of pairs of particle trajectories to indicate when different simulations start to diverge could also be used as in [31]. A comprehensive comparison of the method presented here with other methods is beyond the scope of this paper, but we offer a few thoughts on how this method fits in.

The method has several desirable features for comparison of forecasts. Larger displacements from the reference forecast result in lower scores. Displacement of an area of high concentration will result in a lower score than the same displacement of a low concentration. This results from areas of high concentration being represented by more particles. Identifying and visualizing areas of difference as in Figure 8 is relatively simple. The method takes into account separation in all three spatial dimensions. It is straightforward to implement, particularly if one is already using a GMM for density reconstruction. No thresholds need to be defined.

How much displacement in space lowers the score depends on the concentration gradient. High concentration areas correspond to areas of high probability and a displacement in a direction in which probability changes quickly from high to low will result in a lower score than the same amount of spatial displacement in a direction in which probability changes slowly from high to low. This feature may or may not be desirable depending on the application.

As formulated here, the score is highly dependent on which forecast is used as the reference. $S_{12}$ may be quite different than $S_{21}$ and they should be used in conjunction with each other as well as $S_{11}$ and $S_{22}$.

### 3.3.3. Feature Tracking

GMM are used as clustering algorithms and this aspect can also be used for feature identification. Figure 9 shows an example of feature tracking. A BGMM with 10 Gaussians is fit to the HYSPLIT output every hour for 24 h. The fit from the previous time period is used to create a "warm start" for the next fit. The fitting algorithm identifies features that a human can easily recognize which may be desirable in some applications. The features can be tracked over time. For instance, Figure 9 bottom row shows the progress of the centers of the Gaussians over time. The evolution of the covariance matrix and weights (not shown) give information about which features contain the most mass and how much they are expanding. Air parcel trajectories are often used to derive a big picture of the flow, particularly for emergency preparedness. For instance, NOAA ARL maintains a web page [36] with air parcel trajectories for volcanoes which have a relatively high probability of erupting. The feature tracking graphic shown in Figure 9 bottom row may be able to provide a more accurate picture of the possible evolution of the plume in a similarly compact form as trajectories.

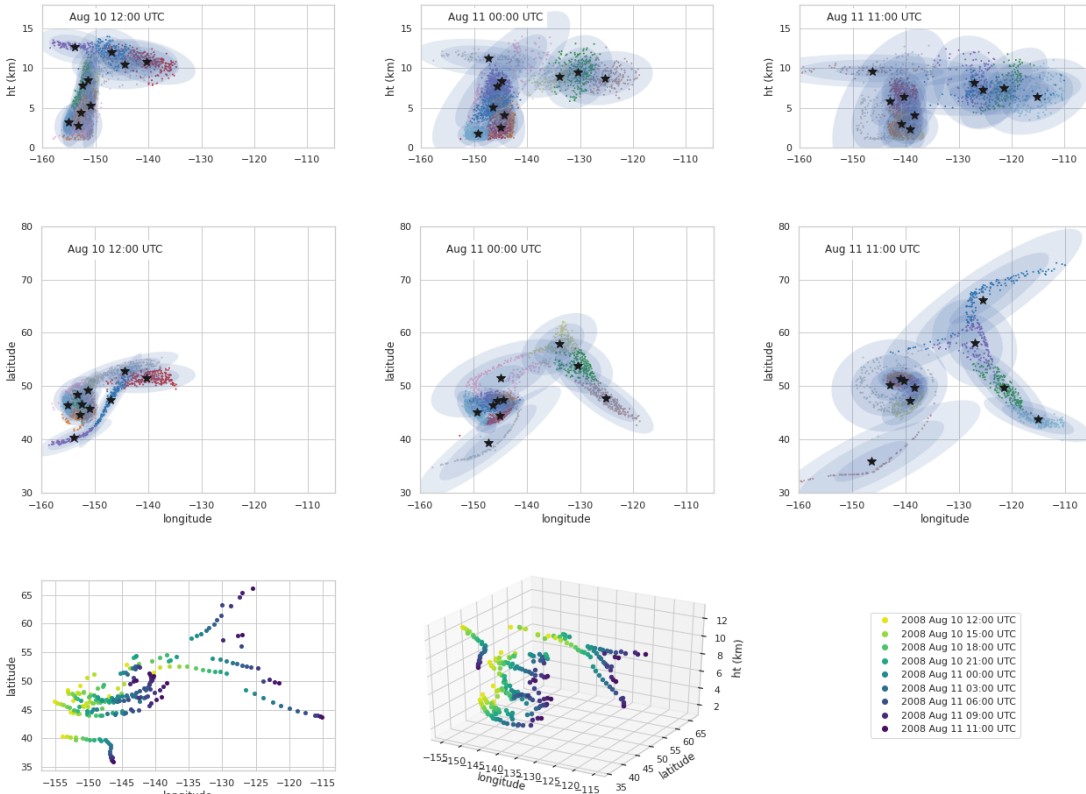

**Figure 9.** The top two rows show features identified by a BGMM with 10 Gaussians at three times. The black stars show the centers of the Gaussians. The colored dots indicate the position of the computational particles. The different colors belong to different groups. The width of each Gaussian is indicated by the blue shaded regions. The top row is a projection onto the height-longitude plane while the second row is a projection onto the latitude-longitude plane. The bottom row shows the position of the centers of the 10 Gaussians every hour from 10 August 12 UTC to 11 August 11 UTC. The plot in the left corner is a projection onto the latitude-longitude plane while the plot on the right is a three dimensional rendering of the positions.

## 4. Discussion

Utilizing a GMM in conjunction with an LPDM opens up a host of possibilities. Reliable density reconstruction can be accomplished with an order of magnitude or more fewer particles. LPDM's are

often used in emergency response applications which require a response within tens of minutes to hours [7] and so computational speed is important.

Currently inversion algorithms can be very computationally expensive as a large number of particles (as many as hundreds of thousands) must be used for each simulation (either forward or backward) and often thousands of simulations must be performed [17,37–40].

Arguably, use of the GMM simplifies the modeling process and makes it more intuitive. Modeled concentrations are significantly less sensitive to the choice of concentration grid and particle number. Nonexpert and even expert users are often stymied by the task of choosing an appropriate grid size and particle number. This can be particularly difficult when a range of spatial and temporal scales is involved. HYSPLIT has evolved many mechanisms to cope with the problem including the use of multiple concentration grids, a concentration grid defined in polar coordinates, and several ways to switch between using computational particles and puffs.

Model output can be stored in a more compact format. The carbon tracker Lagrange [41] provides a database of footprints produced by an LPDM run in backward mode. File sizes for such databases could possibly be reduced significantly by storing only the characteristics of the Gaussian fits, the mean, weight, and covariance matrix for each Guassian, rather than the gridded data.

The original motivation for exploring the GMM was a desire to more fully capture the information conveyed by examining the particle position output of the model and move into a regime where analysis can be performed without a grid. The main output of an LPDM is usually considered to be the gridded concentration data. However, most LPDM's, including HYSPLIT, also output the position (and other characteristics) of the computational particles at predefined output intervals. Visualizations of the computational particle positions are often quite beautiful and fascinating to look at. They tend to convey the three dimensional structure of a complex plume in a manner that is difficult to do with gridded concentration output. Notably, the puff model which was utilized by the Alaskan Volcano Observatory and the Anchorage VAAC for a number of years mostly employed visualizations of the particle positions [42] and this type of output was generally found to be useful to analysts forecasting the position of discernable ash. On the other hand, concentrations, which are usually the relevant quantity, are not conveyed well by such output. Although gridded output is quite convenient for many purposes, an object based approach in which spatial and temporal relationships are more easily preserved and communicated is often more powerful and relatable. The ideas related in Section 3.3 are first steps toward realizing an object oriented analysis of LPDM model output.

The amount of future work that can be performed is considerable. We are eager to combine the GMM density reconstruction with the inversion algorithm in [17]. A rigorous comparison of the GMM density reconstruction with some of the KDE methods should be undertaken. We expect the GMM to be more efficient as a KDE method using a Gaussian kernel still must integrate or store information for as many Gaussians as there are computational particles, while, as demonstrated here, the GMM utilizes considerably less. The number of Gaussians needed is dependent on the area covered by the plume and the complexity of the plume structure rather than the number of particles used in the simulation. Another avenue for future work is determining better criteria for picking the number of Gaussians to be fit. We note that the task may be similar to that of finding the optimal way of defining a KDE, in that many criteria may be good enough. Subjectively we find the GMM approach to be more intuitive than the KDE. Bandwidths which are dependent on particle age or other criteria were developed to help make sure that particles in areas of low particle density have large enough bandwidths to fill the "gaps", while particles in areas of high particle density have smaller bandwidths to avoid excess dispersion. The GMM by contrast uses information about the particle density to create the distribution and so gaps are automatically filled.

There is no particular obstacle to applying the methods described here to simulations requiring much higher spatial and temporal resolution than demonstrated, e.g., urban scales. The method may be particularly useful for smaller scale applications as the number of particles required to achieve an adequate value of $C_\ell$ can be quite high and thus Gaussian Plume models are often preferred at scales

smaller than several km. In cases where topological features need to be resolved, some sort of scheme to handle mass which may be allocated outside of the presumably hard boundary will need to be devised. Mixture models which utilize other distributions besides the Gaussian could also be explored.

Currently the GMM is applied to HYSPLIT model output as postprocessing and no effort has been made to optimize the code. The GMM fitting could be incorporated into the code itself and algorithms for turning the GMM output into gridded data optimized.

All of the methods introduced in Section 3.3 need considerable more investigation and development. The technique described in Section 3.3.2 could be extended for use in dispersion model ensembles, developed further as a method for estimating uncertainty in model predictions and compared to other methods. Additionally, we feel that there are other ways to leverage the mixture models that we have not thought of yet and hope that this work stimulates interest in the subject.

**Supplementary Materials:** The code developed for this project is available from github at https://github.com/noaa-oar-arl/hysplit_gmm. The CONTROL and SETUP.CFG files needed to run the simulations are also included in the repository. We expect that Jupyter notebooks provided with the repository will be helpful for anyone wishing to reproduce or extend the work.

**Funding:** This work was funded and supported by NOAA ARL.

**Acknowledgments:** The author would like to thank Mark Cohen, Ariel Stein, Tianfeng Chai, Fantine Ngan, Barbara Stunder, Allison Ring, Christopher Loughner, and Barry Baker of ARL for modeling support. Fantine Ngan and Chris Loughner provided setup scripts for CAPTEX runs. Barry Baker was helpful in introducing the clustering algorithms available in scipy. Barbara Stunder provided information on how HYSPLIT is run operationally for volcanic ash. Jaime Kibler and Jeff Osiensky at Washington and Anchorage VAACs provided information about VAAC operations. Thank you also to the anonymous reviewers whose insightful comments led to much improved descriptions and discussions.

**Conflicts of Interest:** The author declares no conflict of interest.

## Abbreviations

The following abbreviations are used in this manuscript:

| | |
|---|---|
| ARL | Air Resources Laboratory |
| BGMM | Bayesian Gaussian mixture model |
| CAPTEX | Cross Appalachian Tracer Experiment |
| GMM | Gaussian mixture model |
| HYSPLIT | Hybrid Single-Particle Langrangian Integrated Trajectory model |
| KDE | Kernel density estimator |
| LPDM | Lagrangian Particle Dispersion Model |
| PDF | probability density function |
| n | number of Gaussians used in a fit |
| N | number of computational particles found in a defined volume |
| VAAC | Volcanic Ash Advisory Center |

## Appendix A. Shot Noise

We could not locate a source specifically showing that the number of computational particles in a volume, N, follows a Poisson distribution so we ran both runKB and runB 50 and 100 times respectively with different random seeds and examined the distribution of N in various volumes. We found that the distributions were well represented by a Poisson distribution and three examples are shown in Figure A1.

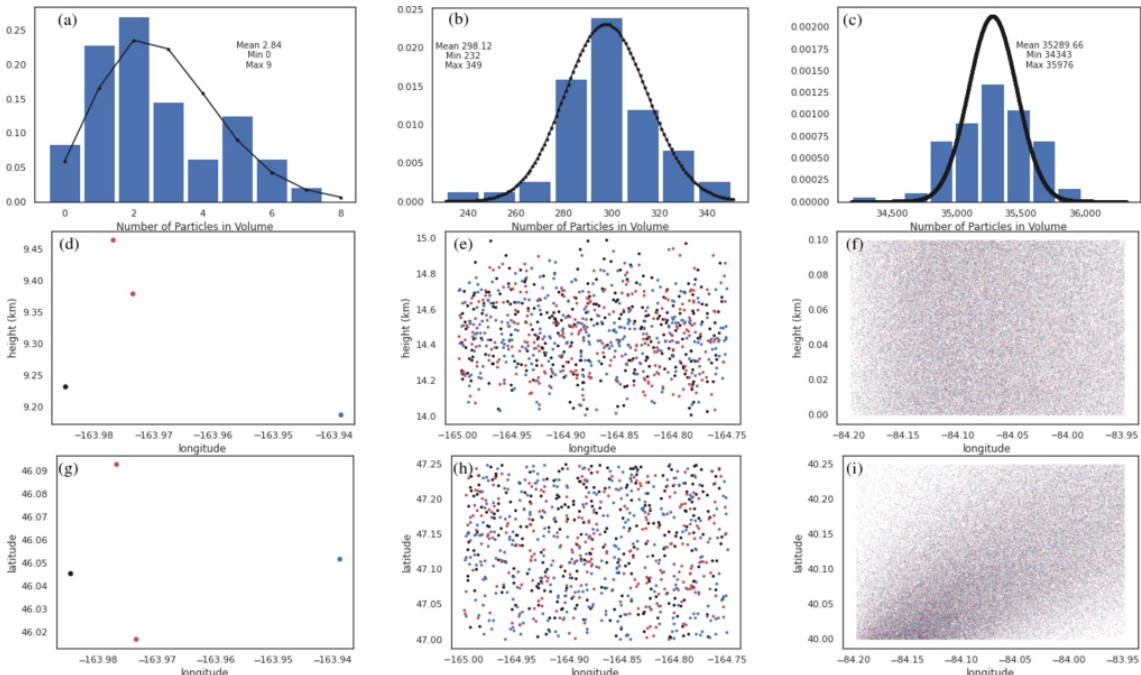

**Figure A1.** Examples of shot noise in the histogram method. The top row (**a–c**) shows normalized histograms (blue bars) of the number of points found in a volume. The black line shows the Poisson distribution with the same mean. The position of computational particles in the volume are shown in the second and third rows. Positions from three of the simulations are plotted with the different colors indicating the computational particles from different simulations. The first and second columns (**a,b,d,e,g,h**) are from 50 simulations of the runKB with output every 10 min from 9 August, 12:10 UTC to 13:00 UTC. The first column (**a,d,g**) looks at a smaller volume ($0.1° \times 0.1° \times 500$ m) and only one particle size while the second column uses a volume of $0.25° \times 0.25° \times 1$ km and all the particle sizes. The last column (**c,f,i**) is from 100 simulations of runB with output every 5 min from 25 September 18:00 UTC to 21:00 UTC and a volume of $0.25° \times 0.25° \times 100$ m. Although the histogram in (**c**) looks somewhat flatter than the Poisson, this may be simply because 100 points is not enough to represent the distribution well.

## Appendix B. Setting the Number of Gaussians

In this section we look at the score $S_{ij}$ as a function of n, the number of Gaussians, for sets of points which were produced by model runs with different seeds. The model runs were identical otherwise and so each set of points should represent the same underlying distribution. The intention is to consider what a good range of n may be. A score is calculated for $n = 5$ through $n = 65$ in intervals of 5. Some examples are shown in Figure A2. The score for $i = j$ generally continues to increase but the rate of increase decreases. This is expected and at some point increasing n leads to overfitting. The score for $i \neq j$, however, levels off at some point and begins to decrease in some cases. Ideally an n which maximizes $S_{ij}$ for $i \neq j$ is desirable. We generally see that a fairly large range of n satisfies this criteria. We only examined these type of plots for a few cases to confirm that our choices of n were within an appropriate range and we leave a more extensive investigation of the most efficient way to determine n for another time.

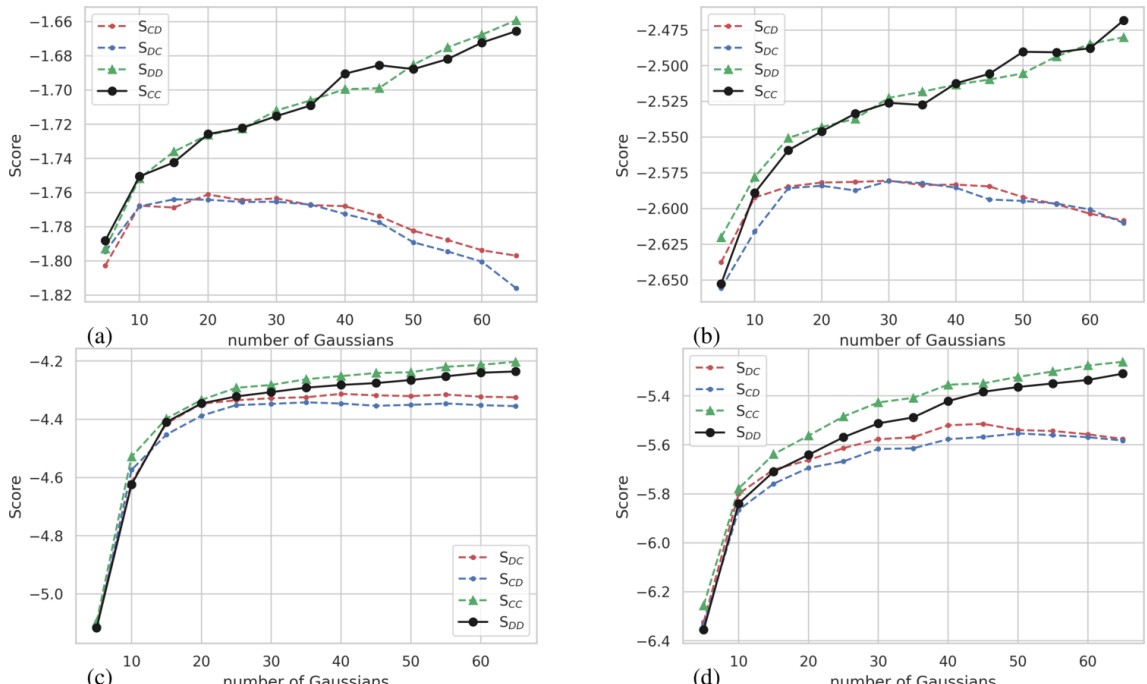

**Figure A2.** Score, $S_{ij}$ as a function of number of Gaussians used in the GMM fit. Black and green lines indicate score for $i = j$ and red and blue lines indicate scores for $i \neq j$. (**a**) RunC and RunD for CAPTEX1 at 09/20/1983 0 UTC. (**b**) RunC and RunD for CAPTEX1 at 09/20/1983 03 UTC. (**c**) RunKC and runKD at 08/09/2008 at 04 UTC. (**d**) RunKC and RunKD at 08/10/2008 12 UTC.

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
