# Peer review of "The Use of Gaussian Mixture Models with Atmospheric Lagrangian Particle Dispersion Models for Density Estimation and Feature Identification"

_atmosphere, doi:10.3390/atmos11121369_

Round 1

Reviewer 1 Report

I have the following comments that the author should carefully consider.

1) The author, rather arbitrarily, decided to use Longitude, Latitude for the horizontal, and  Kilometers for the vertical, to express the particles position in the three-dimensional space for the fitting procedure, thus actually compressing the horizontal scale relatively to the vertical one. Given the results this choice seems sensible, but certainly would not be correct at the Poles. However, the results should not depend on this choice. This seems just a limitation of the algorithm used. How sensitive is the fitting result to this choice, i.e. the coordinates system used?  This should be discussed in detail and it seems as complex as defining the appropriate grid cell size for the considered problem.

2)I found the definition of uncertainty given in equation (4) misleading. With this definition the uncertainty increases with the number of particles, while it is well known that this is not the case. More clearly, it is well known that the statistical error will decrease with the square root of the number of particles used in the simulation.

This misleading behavior stems because (4) is expressed in absolute term and not relative to the concentration that is modelled. This results in a small uncertainty for low concentration. If the definition (4) is normalized by the concentration, the correct behavior of the statistical error decreasing with the square root of the number of particles in the grid cell can be recovered.

3) By reading the manuscript I got the impression that the author used the python library Scikit-Learn as a black box. Even if this is not a manuscript about statistics the author (and the readers) should be well aware of the methods used to fit the Gaussian mixture model to the particles field. The Algorithm selected here and used in Scikit-Learn should be explained in section 2.6 or in an Appendix, and appropriate references to the scientific literature should be included. A detailed description should be given of the score function, eq. 5 (maximized in the fitting algorithm), especially considering that in section 3.3.3, figure 7, a non-standard application of this score is proposed.

4) Why the author did not use also a standard test (e.g. Kolmogorv-Smirnov test) for checking the fitting of the data to the Gaussian mixture model?  

5)Page 8, line 304. The author writes “as time progress, S_14 which describes how well the fit created with the small particle positions fits the large particle positions, remain smaller than S_14…” I think there are two errors here. First, S_14 is always less negative of S_41 (see Figure 7) and this is by definition, since small particles cover a larger domain compared to the coarse particles and the GMM of small particles will envelope the coarse particles distribution in a volume of relatively high probability.

The second error seems a typo, I think that the second S_14 should be S_41.

6) Paragraph 3.3. the author demonstrates that GMM can be used to detect separation of cluster paths. However, this can be accomplished in other ways, especially when the particles members of the clusters are known in advance (as in the current example). This procedure does not seem to have advantages compared to other methods. What is the author opinion?  The author should add a comment about this in the manuscript.

7) line 338, “was more even more…”

8) line 311, Figure 7, should be Figure 8.

Reviewer 2 Report

General comment.
I have found the paper original and very interesting. Even if KDE methods are not a particular novelty and quite commonly used inside LPDMs, methods based on GMM are not so common. The paper shed lights on the positive and also on some of the critical aspects of the method. In general, I would have preferred a little bit more explanation about the GMM method itself inside the paper. In this respect, it appears as a sort of black box throghout the entire paper. An in-depth discussion is not necessary in this context, but a general description of GMM should be useful for the reader, possibly added after paragraph 2.3 describing some features of the Histogram method.
If possible, the author should add some considerations about the time taken by the application of the GMM method derived by the experience in their work. I understand that the python external implementation inside the scikit-learn package has been used, but general considerations in this respect should be useful to better confirm the potential of the method with respect to the other cited ones (KDE and Histogram). Also a brief discussion, at the end of the paper, about the possible extension of the method for applications to other spatial/temporal scales (local and micro) should be useful, taking into account possible difficulties arising such as, for example, the presence of topographical features.

Specific comments and minor corrections along the text.

Page 2, Line 48:
"the underlying distribution is assumed to assume a certain form" please avoid the repetition of "assume"

Page 3, formula number 1:
Please insert a definition of all the parameters used to build RANK, particularly for FMS and KS (R and FB are more commonly used) and an explanation for RANK itself. Draxler [11] is cited as a reference but I have not found any definition there.

Page 3, Line 97:
Please insert a more extensive definition of DT, particles are emitted every DT?

Page 5, Line 182:
Gausian instead of Gaussian

Page 6, Lines 230,231:
The statement "the FB for the GMM is lower than Runs A,B,C,D and comparable to or lower than Run E" seems unclear, is it lower than .... the same indices for the Histogram method?

Page 8, line 303:
"close to the same" maybe is "close to be the same"

Page 8, Lines 304,305:
S14 is repeated, the second one should be S41

Page 8, Lines 328,330:
the statement starting with "a subset of the large particle" is very hard to read and should be rephrased

Page 8, line 339:
In the statement starting with "If the eruption..." the first more should be removed
